# Production of *Purpureocillium lilacinum* and *Pochonia chlamydosporia* by Submerged Liquid Fermentation and Bioactivity against *Tetranychus urticae* and *Heterodera glycines* through Seed Inoculation

**DOI:** 10.3390/jof8050511

**Published:** 2022-05-16

**Authors:** Daniela Milanez Silva, Victor Hugo Moura de Souza, Rafael de Andrade Moral, Italo Delalibera Júnior, Gabriel Moura Mascarin

**Affiliations:** 1Department of Entomology and Acarology, Escola Superior de Agricultura “Luiz de Queiroz”, University of São Paulo (ESALQ-USP), Av. Pádua Dias, 11, C.P. 9, Piracicaba 13418-900, SP, Brazil; danielamilanez@usp.br (D.M.S.); delalibera@usp.br (I.D.J.); 2Crop Science Centre, Department of Plant Sciences, University of Cambridge, Lawrence Weaver Rd., Cambridge CB30 LE, UK; victorhugomour@gmail.com; 3Department of Mathematics and Statistics, Maynooth University, W23 F2H6 Maynooth, Co. Kildare, Ireland; 4Laboratory of Environmental Microbiology, Brazilian Agricultural Research Corporation, Embrapa Environment, Rodovia SP 340, KM 127.5, Jaguariúna 13918-110, SP, Brazil

**Keywords:** biological control, seed treatment, microsclerotia, soybean cyst nematode, two-spotted spider mite

## Abstract

*Pochonia**chlamydosporia* and *Purpureocillium*
*lilacinum* are fungal bioagents used for the sustainable management of plant parasitic nematodes. However, their production through submerged liquid fermentation and their use in seed treatment have been underexplored. Therefore, our goal was to assess the effect of different liquid media on the growth of 40 isolates of *P*. *lilacinum* and two of *P*. *chlamydosporia*. The most promising isolates tested were assessed for plant growth promotion and the control of the two-spotted spider mite (*Tetranychus urticae*) and the soybean cyst nematode (*Heterodera glycines*). Most isolates produced > 10^8^ blastospores mL^−1^ and some isolates produced more than 10^4^ microsclerotia mL^−1^. Microsclerotia of selected isolates were used to inoculate common bean (*Phaseolus vulgaris* L.) seeds in greenhouse trials. All fungal isolates reduced the *T. urticae* fecundity in inoculated plants through seed treatment, while *P. chlamydosporia* ESALQ5406 and *P. lilacinum* ESALQ2593 decreased cyst nematode population. *Purpureocillium lilacinum* was more frequently detected in soil, whereas *P. chlamydosporia* colonized all plant parts. *Pochonia chlamydosporia* ESALQ5406 improved the root development of bean plants. These findings demonstrate the possibility of producing submerged propagules of *P. chlamydosporia* and *P. lilacinum* by liquid culture, and greenhouse trials support the applicability of fungal microsclerotia in seed treatment to control *P. vulgaris* pests.

## 1. Introduction

In a broad sense, biological control seeks to reduce agricultural pests and plant pathogens by means of employing natural enemies, including microorganisms and macroorganisms, in alignment with the minimal risk posed to the environment and human health [1,2]. In 2019, the Brazilian biopesticide market showed a 70% increase in biological products registered for management of agricultural pests, which represented a landmark in the biological control history in Brazil [3]. The microbial control of pests is advantageous due to their narrower host spectrum and higher selectivity than traditional, broad-spectrum chemical pesticides [4]. Many of these beneficial microbes, such as fungi, viruses, protists, and bacteria, are valuable eco-friendly tools for the sustainable regulation or management of insects, mites, and plant–pathogen populations. Furthermore, microbes play an essential role in the mitigation strategy of pest resistance to chemical pesticides and have been ramped up due to the expansion of organic crop production worldwide [4].

Filamentous fungi of the order Hypocreales are recognized as potential biocontrol agents to combat numerous pests. Among those, the soilborne cosmopolitan fungi *Purpureocillium lilacinum* (Hypocreales: Ophiocordycipitaceae) and *Pochonia chlamydosporia* (Hypocreales: Clavicipitaceae) are currently used as microbial pesticides against plant-parasitic nematodes in Brazil [5,6]. Currently, there are 13 registered products against phytonematodes (nine of *P. lilacinum* and four of *P. chlamydosporia*) in Brazil, notably to control *Meloidogyne* spp. and *Heterodera glycines* [7]. Beyond their role as bionematicides, these fungi have also shown insecticidal and acaricidal activities [8,9]. Both *P. chlamydosporia* and *P. lilacinum* are commonly found in a wide variety of soils and naturally inhabit the rhizosphere of both native vegetations and crops. These fungi can produce and secrete extracellular hydrolytic enzymes involved in the colonization process of mature females, cysts, and egg masses of the sedentary plant-parasitic nematodes *Heterodera* spp. and *Meloidogyne* spp. Additionally, both *P. chlamydosporia* and *P. lilacinum* can establish endophytic associations with host plants, thus playing an essential role in plant growth promotion. These features bring desirable outcomes, such as induced resistance and tolerance to biotic and abiotic stresses by modulating host plant phytohormones, biochemical defense responses against the attack of pests and plant pathogens, and alleviation of abiotic stresses and the facilitation of nutrient uptake by colonized host plants [10,11].

In the development of novel biocontrol products, the production of fungal propagules can be accomplished by two main mass production methods: solid and submerged fermentations. In the first, a nutrient-poor substrate, usually with a low water activity, is employed to sustain growth at the interface with the atmosphere, allowing abundant oxygen supply for sporulation, resulting in the aerial conidiophores and conidia. Such asexual propagules consist of important fungal structures for dissemination and infection [12]. This method is easily carried out with low-cost equipment. The major hindrance to this production lies in the time required to produce aerial conidia. Generally, to achieve a high spore yield with this fermentation process takes 10 to 15 days or even longer, depending on the fungus species and isolate. At the same time, it is also dependent on large growth rooms with controlled environmental conditions, such as photoperiod and temperature. Notably, there is trouble in controlling fermentation parameters, such as pH, water activity, aeration, and nutrient levels during fungal growth, while the contamination risk may be more frequent due to the lack of automation.

On the other hand, the submerged liquid fermentation offers a versatile and cost-effective technological platform for various fungal propagules, enabling a shorter cultivation period (i.e., 2–5 days). This technology also allows higher yields, with standardized batch-to-batch uniformity, and greater control of environmental and nutritional growth conditions, in a process with lower contamination risk that is less labor-intensive [12,13]. The propagules produced by submerged liquid culture vary greatly across inter- and intraspecies of fungi. An interesting fungal structure for the pest control is the microsclerotium (hereinafter referred to as MS), a resistance structure that consists of a compact mass of dark-pigmented hyphal aggregates with a size in the range of 50 to 600 µm [14]. Submerged conidia (i.e., microcycle conidia) differ from aerial conidia and blastospores in structural and biochemical features, as the former can be originated from both hyphae or blastospores and shows intermediate hydrophobic and hydrophilic physicochemical characteristics, an adaptation in response to a nutritionally limited growth environment [15,16,17]. Following the penetration of the host through the cuticle, fungal cells are transformed into in vivo blastospores, which are hydrophilic, yeast-like vegetative cells produced by budding, septation, and/or fragmentation of the hyphae within the arthropod hemocoel. These yeast-like cells then continue to propagate by budding and/or fission, in a similar way to what occurs in true yeast cells [18,19,20]. They are also formed during in vitro growth by nutrient-rich and oxygen-rich culture conditions [21]. In contrast, submerged conidia are generally obtained by cultivation under nutritional-poor media [16].

Among the abovementioned fungal structures, there is particular interest in the MS, which are considered a promising alternative to aerial conidia, notably for seed coating and soil inoculation [22,23]. These structures are stable and resilient under field conditions and may remain viable until conditions become more favorable for growth [14]. MS are also more tolerant to adverse abiotic factors, such as UV-B and heat, compared to aerial conidia [24]. The impact of nutritional requirements of liquid media on such submerged propagules has not been investigated yet on the production performance of various isolates from these two nematophagous fungi. Hence, the present work attempts to shed light on the intraspecific phenotypic growth variability in response to varied liquid media compositions for the optimal production of MS and submerged blastospores.

The two-spotted spider mite *Tetranychus urticae* Koch (Acari: Tetranychidae) is among the most polyphagous and cosmopolitan pests of vegetable and legume crops that attacks the phyllosphere of numerous indoor and outdoor cropped plants, including common bean and soybean. Its chemical control is linked with a long history of evolving resistance to acaricides [25,26]. Another economically important global pest is the soybean cyst nematode *Heterodera glycines* Ichinohe (Nematoda: Tylenchida), which stands out as the most devastating parasite of soybean *Glycine max* (L.) Merr. The losses caused by this nematode can be up to one billion dollars for the U.S. economy annually and over 10 billion dollars worldwide [27].

To our knowledge, there is only one study that has investigated the production of MS of *P. lilacinum* by liquid culture along with its in vitro bionematicidal activity against the root-knot nematode *Meloidogyne incognita*, but there are no reports on outcomes from the seed inoculation and their effects on biocontrol activities *in planta* against the two-spotted mite and the soybean cyst nematode, along with growth promotion in common beans. None of these aspects has been explored for *P. chlamydosporia*. In this sense, there is a lack of studies concerning the intra- and interspecific variation in the production of MS and other submerged propagules of these two crucial biocontrol agents by liquid culture coupled to the investigation of seed treatment with MS as a potential inoculation method for inducing control of these two devastating pests from below and above ground, which has motivated the present study. It is also relevant to mention that there are no available products based on either liquid fermentation or the MS of fungal bionematicides worldwide.

In light of current knowledge, no published studies have addressed the application of the MS of *P. chlamydosporia* and *P. lilacinum* to common bean (*Phaseolus vulgaris* L.) seeds and their subsequent effects on the population growth of *T. urticae* and *H. glycines*. Hence, this research aimed to carry out an extensive screening of 42 indigenous Brazilian isolates of *P. chlamydosporia* and *P. lilacinum* by assessing their production of MS and submerged blastospores using different complex liquid media. Additionally, the best MS producers were further selected to assess their endophytic capability in *P. vulgaris* plants through seed inoculation and to further afford plant protection from damages caused by *T. urticae* and *H. glycines* in greenhouse trials.

## 2. Materials and Methods

### 2.1. Source of Fungi and Inoculum Preparation

The isolates used (Table 1) in this work belong to the Entomopathogenic Microorganisms Collection “Prof. Sérgio Batista Alves” of the Laboratory of Pathology and Microbial Control of Insects at the Department of Entomology and Acarology of the University of São Paulo (ESALQ-USP). The selected isolates were identified morphologically and molecularly through sequencing of the nuclear internal transcribed spacer region (ITS) of nuclear DNA with the primers ITS5 and ITS4 [28,29,30]. The evolutionary analysis was performed by a maximum likelihood method with a Kimura 2-parameter model and invariant sites using the software MEGA6 [31,32].

The monosporic cultures from all isolates were made to obtain the inoculum. A standard inoculum used in all bioassays was prepared with aerial conidia, harvested from 10-day-old sporulated cultures grown on potato dextrose agar (PDA, Difco^®^) at 25 ± 1 °C with a 12:12 h (L:D) photoperiod.

The collection, maintenance, and use of these fungal isolates in the present study were conducted following the Brazilian National System of Management of Genetic Heritage and Associated Traditional Knowledge—“SisGen” approved under the protocol RAC856E.

### 2.2. Submerged Liquid Fermentation

Conidial suspensions were obtained from 15-day-old sporulated cultures grown on PDA plates by harvesting conidia with 10 mL of 0.04% aqueous solution of sorbitan monooleate (Tween^®^ 80, Sigma-Aldrich, St. Louis, MO, USA), and then serially diluted to enumerate conidia using an improved Neubauer (hemocytometer) chamber at 400× magnification. All fungal isolates were standardized with aliquots of 5 mL containing 5 × 10^6^ conidia mL^−1^, and were then added into baffled Erlenmeyer flasks (250 mL) (Glass, Bellco^®^, Vineland, NJ, USA) filled with 45 mL of liquid medium, thus adding up to 50 mL per flask. Inoculated liquid cultures were incubated in a rotary shaker with 300 revolutions per minute (rpm) at 28 ± 2 °C (MARCONI^®^, Model: MA 830, Piracicaba, SP, Brazil) in the dark. All culture flasks were shaken manually on a daily basis to prevent mycelial growth on the flask walls.

The liquid medium developed by Iwanicki et al. [33] arose from a combination of two other media described by Mascarin et al. [21] and by Adamék [34], and this was employed in this study to induce the production of blastospores and possibly submerged conidia. The second test medium had a C:N ratio of 10:1 and 36 g carbon L^−1^ (namely Jackson 4), in which the carbon source used was anhydrous glucose (45 g L^−1^, 20% *w*/*v* (40% total carbon), Êxodo Científica Química Fina Indústria e Comércio Ltd., Sumaré, SP, Brazil) supplemented with yeast extract (10% total nitrogen, Difco™, Sparks, MD, USA) as the nitrogen source (45 g L^−1^). The third medium used had a C:N ratio of 50:1 and 36 g carbon L^−1^ (namely Jackson 6, containing glucose 81 g L^−1^ and yeast extract 9 g L^−1^). Both media contained salts, trace metals, and vitamins in the basal medium, which was comprised of KH_2_PO_4_, 4.0 g L^−1^; CaCl_2_•2H_2_O, 0.8 g L^−1^; MgSO_4_•7H_2_O, 0.6 g L^−1^; FeSO_4_•7H_2_O, 0.1 g L^−1^; CoCl_2_•6H_2_O, 0.1 g L^−1^; MnSO_4_•H_2_O, 0.016 g L^−1^; ZnSO_4_•7H_2_O, 0.014 g L^−1^; thiamine, riboflavin, calcium pantothenate, niacin, pyridoxine, thioctic acid, 500 µg L^−1^ each; and folic acid, biotin, vitamin B12, 50 µg L^−1^ each. The modified Adamék medium was prepared with glucose (40 g L^−1^) as a carbon source, yeast extract (80 g L^−1^) and corn steep liquor (40 g L^−1^) as nitrogen sources, and a basal medium containing salts, trace metals, and vitamins as mentioned above. Two, three, and four days after fungal inoculation in the culture media, 1 mL samples were taken under aseptic conditions and were diluted with distilled water + 0.04% Tween^®^ 80 (1:10) to evaluate the concentration of MS on glass slides (25 × 75 mm) (using 50 µL of culture diluted samples) and to examine the culture morphology. Only pigmented MS larger than 50 µm in diameter were counted [14]. For the modified Adamék medium, blastospore concentration was determined for all isolates (see fungal propagules considered in Figure A1).

### 2.3. Drying Fungal Propagules after Submerged Production

Selected isolates were cultivated by submerged fermentation for four days, and then their whole biomass (crude broth) was mixed with diatomaceous earth (DE, 5% *w*/*v*) (Sigma-Aldrich, St. Louis, MO, USA). This mixture was dewatered through filter paper (110 mm diameter, pore size 8 µm, Unifil^®^, Campo Limpo Paulista, SP, Brazil) under vacuum filtration [35]. The final material was transferred to Petri dishes (90 × 15 mm) and submitted to a slow drying process (20 h at 50–60% of relative humidity (R.H.) and an additional 1–2 h at 15–20% R.H.). After reaching water activity ≤ 0.2 measured in the LabMaster-aw equipment (Novasina^®^, Lachen S.Z., Switzerland), samples of 25 mg were immediately used to assess MS viability, and the remainder was vacuum packaged in polyethylene bags and stored at 5 °C until use in further experiments.

For MS viability assessment, 25 mg of MS + diatomaceous earth (MS + DE) was sprinkled on water-agar medium (2% *w*/*v*) in Petri dishes (90 × 15 mm), and then incubated in a growth chamber at 28 °C in the dark. After 24 h of incubation, microsclerotial myceliogenic germination was assessed by randomly counting 100 granules with or without the formation of hyphae under a phase-contrast microscope (DM4000B, Leica^®^ Microsystems, Wetzlar, Germany) [14,35]. The same plates were kept in a growth chamber for an additional seven days to evaluate the sporogenic germination resulting from aerial conidia produced per gram of granules (i.e., conidia g^−1^ of MS + DE). To accomplish this, conidia from sporulated granules on agar-water plates were harvested and serially diluted in 0.04% Tween 80^®^ aqueous solution and were then counted in a Neubauer chamber at 400× magnification.

### 2.4. Seed Treatment of Beans with Nematophagous Fungi, Endophytism, and Growth Promotion

This study sought to assess the endophytic potential and growth promotion in common bean plants through seed treatment using two isolates of *P. chlamyhdosporia* and three isolates of *P. lilacinum*. Initially, seeds of the common bean cv. ‘IAC Milênio’ were surface-sterilized for 45 s in 70% ethanol, followed by 1 min in 1.5% of commercial sodium hypochlorite, for 45 s on 70% ethanol, and washed three times with distilled water [36]. The bean seeds were then coated with MS + DE granules of each selected isolate of *P. lilacinum* (ESALQ1744, ESALQ2482, and ESALQ2593) and *P. chlamydosporia* (ESALQ5405 and ESALQ5406). There was no significant variation between these fungal isolates for spore production by microsclerotial granules (*F* = 0.86, df = 4, 25, *p* = 0.503), and these yields fell in the range of 3.44 ± 0.55 to 4.78 ± 0.55 × 10^9^ (mean ± S.E.) conidia g^−1^ of air-dried MS granules. The MS concentrations were standardized across fungal isolates by estimating the amount of MS required to deliver a field dosage of 10^12^ conidia ha^−1^, according to the conidia production yields obtained by sporulated MS granules upon re-hydration. To improve the adhesion of MS + DE granules to the seed surface, Arabic gum was added at 0.5% (*w*/*w*) to the MS granules during seed coating at a rate equivalent to 400 g Arabic gum to 100 kg of seeds. For this purpose, the mixture with Arabic gum was hydrated with sterile distilled water (approx. 150–200 µL mixed with 0.135–0.140 g of Arabic gum) for better adherence to the seeds. The control consisted of only common bean seeds treated with Arabic gum. Treated seeds were placed on a filter paper (110 mm diameter, pore size 8 µm, Whatman^®^, Little Chalfont, UK) to dry out the excess water, then these seeds (ten seeds per treatment) were sown in 1-L plastic pots filled with non-sterile (natural) potting mixture containing approx. 1.2 kg of 40% sand and 60% soil. Potted bean plants were kept in a greenhouse (22°42′46.4″ S; 47°37′36.3″ W) and were watered daily with tap water if necessary.

The entire experiment was repeated three times on different occasions under greenhouse conditions. A randomized block design was used with ten biological replicates (e.g., potted plants) for each treatment and a total of six treatments (control and five fungal treatments). An illustrative schematic procedure of seed treatment and the experimental setup is available in Figure A2.

At the end of the trial (45 days after fungal inoculation and seeding), the plants were removed from the pots, and their roots were washed to remove the soil. Measurements of the height of the aerial part (cm), root length (cm), and dry weights (g) of potted bean plants were carried out. In parallel to that, three fragments of roots, stems, and leaves were taken from each 45-day-old potted plant to assess the fungal colonization ability within the host plant tissues (i.e., endophytism). Pieces of plant parts were surface-sterilized by means of the following steps: immersion in 70% ethanol for 30 s, 1.25% commercial sodium hypochlorite for 60 s, 70% ethanol for 30 s. They were then rinsed in sterile distilled water three times before plating. Subsequently, aliquots of 50 µL of the last rinsing water were collected at the beginning and end of the sterilization process and were plated on PDA medium to confirm the success of the sterilization [37]. Surface-sterilized fragments of roots, stems, and leaves were placed on sterile filter papers to remove the excess water. These plant pieces were then plated on PDA medium amended with 0.5 g L^−1^ of cycloheximide, 0.2 g L^−1^ of chloramphenicol, 0.5 g L^−1^ of Dodine (65% *w*/*v* Dodex^®^ 450 SC, Sipcam Nichino Brasil S/A, Uberaba, MG, Brazil), and 0.01 g L^−1^ of crystal violet (Dinâmica Química Contemporânea & Reagentes, Indaiatuba, SP, Brazil) [37]. These inoculated agar plates were incubated in a growth chamber for 20 days until fungal outgrowth was evident from those plant pieces. Fungal colonization was visually examined based on colony morphology under a microscope to identify fungal structures. For the soil analysis, 1 g of soil was diluted in 10 mL of distilled water + 0.05% Tween^®^ 80, and then 50 µL aliquot of the dilutions 10^−1^, 10^−2^, and 10^−3^ were plated on the same selective medium and incubated at 25 ± 1 °C, 12:12 h L:D for 15 days. The colonies were morphologically examined under a phase-contrast microscope to identify both species of nematophagous fungi tested in this experiment.

### 2.5. Effects of Nematophagous Fungi on the Population Growth of the Two-Spotted Spider Mite

In this part of the study, the impact of fungal endophytes on the population growth of the two-spotted spider mite was evaluated in bean plants whose seeds were inoculated with air-dried MS granules of five selected nematophagous fungal isolates (seed treatment described in Section 2.4.) One transparent plastic clip cage (4.5 cm high × 3.8 cm diameter) with a mesh at the top (0.09 mm mesh size) was placed on each bean plant in the V4 stage, approximately 21 days after fungal inoculation and seeding [38]. Subsequently, one *T. urticae* female was individually transferred to each clip cage. The spider mite fecundity was estimated on the 7th day of infestation by counting the number of eggs, and eventually, any larvae emerged and were thus expressed as the number of eggs produced by mite or fecundity, as a means to measure the population growth of this mite. Treatments were set up using a randomized complete block design with ten plants (biological replicates) per treatment. The control group was represented by fungus-free bean plants only infested with the spider mite. The entire experiment was independently repeated three times using new fungal inoculums freshly obtained from different production and drying batches.

### 2.6. Effects of Nematophagous Fungi on the Soybean Cyst Nematode

The *H. glycines* ‘race 3’ populations used in the greenhouse experiments were provided by the Laboratory of Nematology, located at the Department of Plant Pathology and Nematology (ESALQ/USP). The inoculum consisted of a suspension containing 1000 individuals (eggs + second-stage juveniles [J2]) and was obtained from soybean plant roots, which were previously inoculated and maintained in a greenhouse. The *H. glycines* cysts were extracted following the method proposed by Machado and Silva [39]. They were manually ground in a 500-mesh sieve using a rubber and were then collected and transferred to a beaker.

Prior to the inoculation, the nematode suspension was homogenized using a magnetic stirrer. Each experimental unit (1-Lplastic pots filled with approx. 1.2 kg of natural (non-sterile) 40% sand and 60% soil potting mixture containing a common bean plant at V4—approx. 15 days after seeding) was inoculated with the nematode suspension by pouring it into two holes (2 cm and 4 cm depth) close to the plant stem. Vermiculite was added to the holes to avoid heat stress, and after the inoculation, the plants were transferred to the greenhouse until the evaluation.

The evaluation process occurred 45 days after the fungal inoculation and seeding [40]. The nematodes were extracted following the methodology described by Coolen and D’Herde [41]. The bean plants were removed from the pots, and the roots were carefully washed and cut into 1 cm pieces. The fragments were blended using a 1% commercial sodium hypochlorite to dissolve the eventual egg masses. The obtained nematode suspension was poured through two sieves, the first of 60 mesh (0.260 mm—to retain the coarse particles) and the second of 500 mesh (0.025 mm—to keep fine particles). The nematodes were collected in a beaker and centrifuged twice. In the first one, kaolin was added to each sample, followed by centrifugation at 1800 rpm for 5 min. In the second, the supernatant was discarded, and a sucrose solution (1.15 g cm^−3^) was added, followed by centrifugation at 1800 rpm for 1 min. The nematodes were collected using a 500 mesh sieve, and were then killed in low heat (approx. 50–60 °C) and fixed by adding 300 µL of formaldehyde. Samples were stored in the fridge (8 °C) until further analysis.

The number of nematodes was estimated by performing two 0.5 mL counts on a Peters’ counting slide under an optical microscope. After that, the final population and nematodes per gram of root were estimated from the total fresh root mass collected in each potted plant. A randomized block design was used with ten biological replicates (e.g., potted plants) for each treatment and a total of six treatments (control and five fungal treatments). This study was carried out in two independent greenhouse trials set up on different occasions using freshly produced fungal and nematode inoculums.

### 2.7. Data Analysis

The microsclerotia production of *P. lilacinum* and *P. chlamydosporia* were analyzed using Poisson generalized linear mixed models (GLMMs), including the effects of the experiment, isolate, day, medium, and the two and three-way interactions between isolate, day, and medium as fixed, as well as random day effects per experimental unit, medium and isolate combination, and an observational-level random effect of accounting for overdispersion. A linear mixed model was fitted to the logarithm of the submerged propagules yield, including the effects of the experiment, isolate, day, and the interaction between isolate and day as fixed and random day effects per experimental unit in the linear predictor. In this case, we also modeled the variance as a function of isolate to account for variance heterogeneity. The significance of the effects was assessed using likelihood-ratio tests at *p* < 0.05. As a complementary approach, we carried out an analysis that allowed us to look at the marginal performance of fungal isolates as random effects, regardless of medium and day of fermentation, in which we were able to establish a ranking order from the best to the weakest producers of MS and submerged blastospores.

Hierarchical clustering analysis was performed using microsclerotia and submerged propagules data for multiple fungal isolates. Predicted means derived from model fits were normalized before obtaining the Euclidean distances, a similarity measure. Subsequently, the distances were clustered, taking into account measurements for liquid media and fungal isolates using Ward’s method. We then produced a heat map using the ‘gplots’ package [42].

The female fecundity (number of eggs laid by each spider mite female up to seven days upon feeding on bean leaves) data were analyzed by fitting a Poisson model using the log link function. The additive effects of experiment and treatment were included in the linear predictor. The significance of the effects was assessed using Chi-squared tests for the difference between deviances. Multiple comparisons were performed by obtaining the 95% confidence intervals for the estimated treatment effects and the Tukey HSD method using the package ‘emmeans’ [43].

The soybean cyst nematode final population and the number of nematodes per gram of fresh root mass were fitted to generalized additive models (GAMS) with Gumbel distribution for errors and log-link function and included the isolate effect in the linear predictor [44]. When a treatment effect was found to be significant, then a likelihood ratio test was employed to compare each fungal treatment directly with the untreated control group (*p* < 0.05) to check whether the fungus was able to reduce the nematode population growth.

All plant trait variables (plant responses to fungal isolates) were analyzed independently for each experiment performed under greenhouse conditions using linear mixed models, including fungal isolate as a fixed effect and potted plant as the random effect in the linear predictor. The response variables recorded after 45 days of bean plant growth in the greenhouse encompassed aerial root length, plant height, whole plant dry weight, root, and foliage dry weight. The significance of the effects was assessed using *F* tests. Significant differences between treatments were determined by multiple comparisons based on contrast using the Tukey HSD method with 5% of significance with the package ‘emmeans’ [43].

Goodness-of-fit for model selection was assessed using half-normal plots with a simulated envelope [45] and wormplots [44]. All analyses were carried out using R [46].

## 3. Results

### 3.1. Microsclerotia (MS) Production

The majority of *P. lilacinum* and *P. chlamydosporia* isolates produced MS when cultivated in three different liquid culture media: modified Adamék, Jackson 6 (C:N ratio 50:1), and Jackson 4 (C:N 10:1). The exception here was noted for isolates ESALQ1907 (did not produce on Jackson 4 and 6), ESALQ1908 (did not produce on modified Adamék), and ESALQ2509 (did not produce on modified Adamék and Jackson 4) (Figure 1A). The mature microsclerotia from four-day-old cultures exhibited darker pigmentation when grown in the modified Adamék medium than the other tested media, as the former is richer in nitrogen and iron sulfate.

The MS production differed among culture media, and the concentration also varied among isolates and cultivation time (Figure 1B–D). For the Jackson 4 medium, the production ranged from 2 × 10^3^ to 1.7 × 10^4^ MS mL^−1^; and the most productive isolates in this medium were ESALQ1906, ESALQ1909, ESALQ2164, ESALQ2166, ESALQ2482, ESALQ2593, ESALQ2599 and ESALQ2765 (1.0–1.70 × 10^4^ MS mL^−1^), and the ones with the lowest MS mL^−1^ concentration was attributed to ESALQ668 and ESALQ2832 (2 × 10^3^ MS mL^−1^).

For the Jackson 6 medium, production ranged between 3 × 10^2^ and 6 × 10^4^ MS mL^−1^, with the most productive isolates being ESALQ1744, ESALQ1771, ESALQ2164, ESALQ2599, ESALQ2645 (1.0–1.96 × 10^4^ MS mL^−1^), ESALQ2482 (6.1 × 10^4^ MS mL^−1^) and ESALQ2593 (2.2 × 10^4^ MS mL^−1^). On the other hand, the isolate ESALQ5406 was ranked the weakest producer (3.3 × 10^2^ MS mL^−1^).

In the modified Adamék medium, yields of MS ranged from 1 × 10^3^ to 2 × 10^4^ MS mL^−1^ on the fourth day of fermentation, where five isolates stood out as the most productive ones: ESALQ2078, ESALQ2164, ESALQ 2509, ESALQ2593, and ESALQ2718 (1.0–1.16 × 10^4^ MS mL^−1^). Conversely, the lowest numbers of MS were attained by isolates ESALQ668, ESALQ1996 and ESALQ2776 (1.66 × 10^3^ MS mL^−1^). Besides that, isolate ESALQ5406 appeared to be recalcitrant when grown in this medium, as it did not produce any MS.

According to the estimated general mean of the ranking graph (Figure 1E) for fungal isolates based on all media and fermentation time, the three isolates with the highest rank that stood out were: ESALQ2482, ESALQ2164, and ESALQ2593 of *P. lilacinum*. On the other hand, the three isolates considered the least productive were ESALQ5406, ESALQ5405, and ESALQ2776. In comparison to *P. lilacinum*, *P. chlamydosporia* isolates (ESALQ5406 and ESALQ5405) did not perform well regarding MS production in any culture media, which reflected in productions ranging from 3.3 × 10^2^ to 7 × 10^3^ MS mL^−1^. All MS formed by these nematophagous fungi by day four of cultivation appeared well-formed and pigmented, depicting a compact hyphal aggregation within the size range of 50–600 µm (Figure 1F).

### 3.2. Submerged Blastospore Production

In the modified Adamék medium, the concentration of blastospore-like propagules was observed in larger numbers that increased over time in all fungal isolates (Figure 2A,B). Looking at the marginal performance of isolates corrected by experiment and fermentation day (Figure 2B,C), it was possible to rank the strongest to weakest producers of submerged propagules among the isolates tested. In this regard, the two *P. chlamydosporia* isolates, ESALQ5405 followed by ESALQ5406, showed the highest production yields of blastospores, while the weakest producer was the isolate *P. lilacinum* ESALQ1909. These results also corroborate the heatmap output and facilitate the selection of the best fungal isolates. Interestingly, the production of submerged conidia for these fungal isolates grown in any of the three liquid media tested here was not observed.

### 3.3. Effect of Nematophagous Fungi on Bean Growth through Seed Treatment

There was a significant variation among greenhouse trials regarding the plant traits evaluated after 45 days of growth. Several characteristics did not differ statistically from control plants, which include the mass of aerial portion (Trials 1, 2, and 3), total plant mass (Trials 1, 2, and 3), and plant length (Trials 1, 2, and 3) (Figure 3). However, differences among the strains are evident (Figure 3). For example, no statistical differences were obtained in length compared to control plants, but ESALQ2482 differed from ESALQ2593 (Trial 1).

A decrease in the weight of the aerial part of the plants was observed for *P. chlamydosporia* isolates in trial 2 (*p* < 0.05), but the isolates of *P. lilacinum* were grouped with the control (Figure 3). In the same trial, plants treated with *P. chlamydosporia* presented a lower weight than plants treated with *P. lilacinum* ESALQ1744 (Figure 3).

An increase in root mass was obtained in trial 2 (Figure 3), in which all isolates of *P. lilacinum* and *P. chlamydosporia* differed statistically from the control (*p* < 0.05). In trial 3, all isolates were statistically similar to the control, although variability can be observed among the strains. Furthermore, it is important to highlight a tendency of root mass increasing of ESALQ1744 and ESALQ5406, which differed statistically from isolates ESALQ2593 and ESALQ5404, but not from the control plants.

Regarding plant height, no statistical difference was obtained in trial 1 and trial 2. In trial 3, an increase was obtained in plants treated with *P. lilacinum* ESALQ2593, which differed statistically from the control (*p* < 0.05). In the same trial, a trend of increased height was observed for the other isolates, but they did not differ statistically from both control and ESALQ2593.

### 3.4. Fungal Colonization in Potted Bean Plants

The endophytic ability of these nematophagous fungal isolates in potted bean plants was confirmed after seed inoculation with their MS. The spatial distribution of this endophytic relationship between fungal isolates and bean plants was measured as the incidence of the fungus in different plant parts. The persistence of these isolates in the potting soil was also determined by the incidence of the fungus in soil samples. There was a strong interaction between fungal isolate and substrate (plant parts and soil) on the incidence of fungal colonization from the bottom-up in potted bean plants (*χ*^2^ = 100.88, df = 15, *p* < 0.0001). In that way, the five fungal isolates (three of *P. lilacinum* and two of *P. chlamydosporia*) were able at different levels to endophytically colonize different parts of bean plants. All five fungal isolates colonized bean roots (10–52.9%), but only isolate *P. chlamydosporia* ESALQ5405 was not retrieved from the potting soil (Figure 4A). No nematophagous fungi were recovered from untreated control plants, even though the soil was not sterile, indicating the absence of natural resident endophytic fungi. Notably, the highest incidence of endophytism was detected in bean roots, followed by stem (10–47.06%) and leaf (0–57.1%). The three isolates of *P. lilacinum* were recovered from the stems and roots of bean plants, and they were also more frequently detected in potting soil than the two isolates of *P. chlamydosporia* (Figure 4A,B). In contrast, both isolates of *P. chlamydosporia* were either null or detected with low frequency in the soil (Figure 4A). Interestingly, *P. chlamydosporia* ESALQ5405 was the only fungus retrieved from leaf fragments and capable of colonizing all plant parts and the soil, whereas the other isolate—ESALQ5406—was only detected in roots and stems. These results indicate phenotypic variations in the spatial distribution of plant endophytism and soil competence among these fungal isolates.

### 3.5. Effects of Nematophagous Fungi on Population Growth of T. urticae

The effect of fungal isolates through seed coating in grown common bean plants exposed to two-spotted spider mites was measured by the production of eggs per leaf. As shown in Figure 5, the indirect acaricidal effect exerted by all *P. chlamydosporia* and *P. lilacinum* isolates was significant, as these fungi induced a striking reduction in the egg numbers laid by spider mite females (*χ*^2^ = 257.97, df = 5, 113, *p* < 0.0001). No differences were observed among these fungal isolates, which implies that they were similarly effective in suppressing *T. urticae* offspring (Figure 5).

### 3.6. Effects of Nematophagous Fungi on the Population Growth of H. glycines

Two independent greenhouse trials were carried out to assess the impact of isolates of two nematophagous fungal species against *H. glycines* in potted bean plants. Since these two experiments were conducted at different periods of the year, they greatly varied in terms of temperature, relative humidity, and photoperiod, and also exhibited a significant difference in magnitude between trials regarding the final nematode population in untreated plants (Trial 1: 507 nematodes versus Trial 2: 142 nematodes). Regarding the *H. glycines* final population, there was a significant effect in decreasing the nematode numbers displayed by isolates *P. chlamydosporia* ESALQ5406 and *P. lilacinum* ESALQ2593 when compared to control plants infested only with *H. glycines* in the first (*χ*^2^ = 12.75, df = 5, *p* = 0.0258) and the second trial (*χ*^2^ = 15.34, df = 5, *p* = 0.0090) (Figure 6). Consistently, these two nematophagous fungal isolates significantly reduced the final nematode population by approximately 2-fold and 3.5-fold compared to untreated plants infested with *H. glycines*, respectively. The other fungal isolates did not have a meaningful effect on the reduction of the final nematode population. Interestingly, bean plants treated with the isolate *P. lilacinum* ESALQ1744 portrayed a similar trend to that found in untreated plants. Actually, untreated plants had an increase in population growth of *H. glycines*, while *P. lilacinum* ESALQ1744 was unsuccessful in reducing nematode reproduction in both trials.

Regarding the number of nematodes per gram of plants, all isolates of both fungal species reduced considerably the nematode density compared to control plants infested only with *H. glycines* in the first trial (*χ*^2^ = 29.78, df = 5, *p* = 0.00002). In the second trial, both isolates of *P. chlamydosporia* (ESALQ5406 and ESALQ5405) and one isolate of *P. lilacinum* (ESALQ2593) displayed a strong suppression effect in nematode density in comparison to uninoculated plants (*χ*^2^ = 13.64, df = 5, *p* = 0.01808) (Figure 7). Notably, in the first trial, when infested plants from untreated control exhibited higher population density of *H. glycines* (e.g., 1061 nematodes g^−1^ of roots), all nematophagous fungal isolates were capable of suppressing this parasite growth by 2.6–7.3-fold in comparison to untreated plants. The lowest nematode density was attained by *P. lilacinum* ESALQ2593 (e.g., 146 nematodes g^−1^ of roots). In the second trial, both isolates of *P. chlamydosporia* (e.g., 25.1–25.3 nematodes g^−1^ of roots) and the isolate of *P. lilacinum* ESALQ2593 (e.g., 36.2 nematodes g^−1^ of roots) were consistently effective at reducing nematode population density in bean roots by a magnitude of 1.2–2.8-fold lower nematodes than in untreated plants (e.g., 70.1 nematodes g^−1^ of roots).

## 4. Discussion

The present study shows the potential of producing novel propagules of two key nematophagous fungi—*P. chlamydosporia* and *P. lilacinum*—by submerged liquid fermentation and to explore them for the biocontrol of economically important agricultural pest species. The potential of these fungi to control plant–parasitic nematodes is well known, but here we revealed their potential to also manage spider mites. Despite their phylogenetic distance and multiple ecological roles, both fungi are widely recognized as plant-growing promoting fungi (PGPF), in which they can directly act against plant pathogens or increase plant fitness by colonizing the root system [47,48,49]. Often, this colonization by endophytic fungi results in growth promotion, nutrient uptake, and induced resistance against biotic and abiotic stresses [50,51]. Therefore, acquiring high-quality propagules through a cost-effective mass production system is imperative to achieve multifunctional fungal biopesticides, once they are valuable additional tools in sustainable crop pest management programs.

Liquid fermentation provides many advantages when compared to traditional solid fermentation, including labor-saving, less time-consumption, and more controllable growth conditions [12,13]. In our results, 0.1–1.0 × 10^5^ MS mL^−1^ and 1.0–1.2 × 10^9^ yeast-like blastospores mL^−1^ were obtained by the best isolates of *P. lilacinum* and *P. chlamydosporia* within only four days of cultivation. To the best of our knowledge, this is the first comprehensive study investigating the impact of different liquid media on several isolates of *P. lilacinum* and *P. chlamydosporia* for the production of blastospores and MS by liquid culture. In agreement with our results, Song et al. [52] were the first to describe the ability of *P. lilacinum* to form persistent MS and the indispensable role of iron in its formation during liquid cultivation, and further showed that this type of propagule attains excellent nematophagous ability and greater thermotolerance and UV-B radiation tolerance compared to aerial conidia produced by solid substrate fermentation. However, the production of blastospores by *P. lilacinum* in liquid culture remains underexplored up to now. Microsclerotia (MS) and blastospores of *P. chlamydosporia* are preferred propagules to replace aerial conidia or chlamydospores. The results shown here open opportunities to explore submerged propagules of both species of nematophagous fungi for crop protection and growth promotion in agricultural production systems that may contribute to a reduction in reliance on chemical inputs.

A mass production through cost-effective and efficient submerged liquid fermentation of nematophagous fungi represents a groundbreaking advance for making these biocontrol agents economically viable and industrially scalable at a competitive cost compared to conventional solid-state fermentation currently practiced in developed and developing countries. To illustrate the relevance and demand for biological nematicides, most of the microbial biopesticides registered in Brazil in 2021 have been targeted at controlling plant-parasitic nematodes [7]. Hence, the submerged liquid fermentation may facilitate the scale-up while minimizing operational costs for the mass production of resilient propagules of these nematophagous fungal species, which could be well-suited to seed and soil applications.

Microsclerotia (MS) are dark-pigmented overwintering structures produced by some fungi under certain stress conditions and can be induced by submerged liquid cultivation. The MS exhibit a remarkable tolerance to abiotic/biotic stresses and excellent storability [53,54]. These traits make them very promising to agriculture, and their growth and nutritional requirements have been extensively studied, notably in *Beauveria* and *Metarhizium* [14,55,56,57]. In our work, the highest production of MS was achieved with high concentrations of carbon, with the C:N ratio of 50:1 (Jackson 6 medium), which is in agreement with the optimal production obtained for *Trichoderma harzianum*, with carbon content >25 g L^−1^ and C:N ratio from 10:1 to 50:1 [35]. The MS production seems to be triggered by the depletion of nitrogen, while carbon consumption induces the melanization process [14]. A lower C:N ratio led to more melanized MS at the end of the fermentation period (culture media Jackson 4 with 10:1 and modified Adamék in this study), although fewer were produced. Song et al. [54] observed that when culturing *Metarhizium rileyi* (formerly *Nomuraea rileyi*) in nitrogen and carbon-deficient liquid medium, no MS were formed, suggesting that a combination of the two nutrients is essential in the formation of this resting structure.

Microsclerotia (MS) comprise an alternative active ingredient to aerial conidia or environmentally sensitive submerged blastospores. The former is a resting fungal structure that may persist longer in soil than the latter. Generally, the soil antifungal activity can limit the persistence and survival of nematophagous fungi, thus influencing nematode control. Accordingly, propagules of *P. chlamydosporia* or *P. lilacinum* can be inhibited from germinating due to soil antifungal activity [58]. Microsclerotia (MS) of both nematophagous fungal species may be formulated for seed and soil treatments against plant-parasitic nematodes, and they may account for long persistence in the soil due to their natural resilience.

Blastospores are yeast-like vegetative single cells that are initially produced by hyphae in many situations, e.g., in the arthropod’s hemolymph [20]. This propagule can be produced in high concentrations by rich carbon-nitrogen liquid media and has the advantage of having a faster germination pace than aerial conidia [17,59,60]. Several biological control microorganisms can produce such cells, including *Metarhizium* [61], *Beauveria*, and *Cordyceps* [21,62,63]. To our knowledge, the production of these structures by *P. lilacinum* and *P. chlamydosporia* has not yet been reported. The production of submerged blastospores are greatly dependent on the concentrations of carbon and nitrogen in the liquid medium [16,18], which were present in the liquid media tested in this work. Romero-Rangel et al. [64] obtained 3.8 × 10^7^ blastospores mL^−1^ of *Hirsutella* sp. in seven days; Jackson et al. [65] reported the production of 1 × 10^9^ blastospores mL^−1^ for *Cordyceps fumosorosea* with four days of growth, changing only the nitrogen source. In contrast, the highest concentrations were achieved with the optimization of the submerged culture conditions by Mascarin et al. [21,62] Compared to earlier studies, our work assessed 42 isolates, of which some of them were able to produce blastospores at yields of up to 10^9^ mL^−1^ in four days of cultivation, which would be expected for fast-growing propagules. The manipulation of the liquid medium plays a pivotal role in acquiring high concentrations of the desired propagule within shorter fermentation times and with the ability to endure osmotic/oxidative stress during desiccation [12]. It is therefore important to note that for the development and growth of these two fungal species, their different isolates may require different optimum nutritional and environmental conditions during submerged liquid fermentation. To this end, it is worthwhile to conduct additional research in order to understand better the nutritional and ecological requirements case-by-case for each fungal isolate.

In our research, common bean plants inoculated with *P. lilacinum* ESALQ1744, ESALQ2482, and ESALQ2593, and *P. chlamydosporia* ESALQ5405 and ESALQ5406 decreased the fecundity and, consequently, the population growth of the *T. urticae* population. Corroborating this result, Canassa et al. [38] reported suppression in the population growth of *T. urticae* after treating seeds of *P. vulgaris* with *M. robertsii*, *B. bassiana,* or both. The treatment of cotton seeds with *P. lilacinum* and *B. bassiana* decreased the survival of *Helicoverpa zea* [66]. Colonization of PGPFs could trigger the latent defense mechanisms within the plants, leading to systemic acquired resistance or induced systemic resistance, which are mediated by salicylic acid and jasmonic acid/ethylene, respectively [49,50]. Sometimes, plants undergo a state of “priming”, in which they are in a “state of alert”, and the mechanisms of resistance are more intensely expressed with the arrival of the stressor, and in a short time lapse [50]. However, the effect of endophytic (more specifically, the entomopathogenic fungi) is not limited to it. Notably, *B. bassiana* colonization in plants infected with *Exserohilum turcicum* increased the relative abundance of beneficial bacteria, such as *Burkholderia* and *Pseudomonas* [67]. The authors also reported that the microbiome complexity increased after treatment with *B. bassiana*. Recent studies have shown that soil and plant microbiomes play an important role in plant resilience against different sources of stress [68]. Therefore, it is noteworthy to highlight the importance of soil and plant microbiomes and the need for further investigations on the interactions between soil-plant microbiomes and endophytic colonization displayed by *P. lilacinum* and *P. chlamydosporia* in plant resilience and health challenged by abiotic and biotic stresses.

Our data revealed that some isolates of the nematophagous fungi tested were able to improve bean growth, while others were consistently effective in suppressing the cyst nematode population growth in bean plants. Particularly, the *P. chlamydosporia* isolate ESALQ5406 promoted a clear and pronounced growth effect on bean plants by boosting root development, and also proved to be a good endophyte. More interestingly, all nematophagous fungal isolates selected in this study strongly impaired the fecundity of *T. urticae* on the phyllosphere, which indicates that their endophytic association with bean plants can effectively trigger defense responses to aboveground attacks from phytophagous mites. To the best of our knowledge, this is the first time that isolates of *P. lilacinum* and *P. chlamydosporia* are reported to decrease the spider mite population growth in *P. vulgaris* plants whose seeds were inoculated with their MS. In the context of this tri-trophic interaction, our results tie well with previous studies, showing that seed inoculation with entomopathogenic fungal endophytes reduced spider mite population growth in bean plants [38,69]. This outcome is possibly related to defense responses against herbivores triggered by fungus-inoculated plants (e.g., proteins and secondary metabolites) and/or by the production of fungal secondary metabolites in planta [70,71]. Hence, it is possible that fungus-inoculated bean plants can either synthetize and accumulate toxic compounds to mites in leaf tissues, or that the fungus can also secrete internally acaricidal compounds that pose harmful effects on the mites. A snapshot of the molecular interactions of the endophytic growth of *B. bassiana* within the common bean plant (*P. vulgaris*) infested with *T. urticae* revealed the involvement of key fungal genes expressing degrading enzymes and sugar transporters [72]. Overall, the underlying biochemical and molecular mechanisms of these plant–fungus–mite and plant–fungus–nematode interactions require further in-depth investigations.

Regarding the *H. glycines* trials, the relatively low reproduction (below 1000 individuals/plant) could be due to the natural egg diapause, as previously described [73,74]. The method employed in this study for the extraction of nematode eggs has the advantage of retrieving a great number of eggs, but some of them may be in diapause. Nevertheless, in both experiments, we observed a decrease in the *H. glycines* reproduction by reducing the total number of nematodes per plant or gram of root. Both *P. lilacinum* and *P. chlamydosporia* are distinguished nematophagous fungi, as they are able to parasitize eggs, second-stage juveniles, mature females of sedentary nematodes, and cysts (e.g., *Meloidogyne* spp. and *Heterodera* spp.) [50,75,76] In addition, these fungi can trigger defense-related genes in the host and thus suppress plant-parasitic nematodes [58].

Only two fungal isolates, e.g., *P. chlamydosporia* ESALQ5406 and *P. lilacinum* ESALQ2593, consistently decreased the cyst nematode population in all greenhouse trials. However, it is noteworthy that all isolates (trial 1) and isolates ESALQ5405, ESALQ5406, and ESALQ2593 (Trial 2) diminished the number of nematodes per gram of root (Figure 7). In other words, although the final population is the same, proportionally (based on gram of roots), plants presented fewer nematodes due to a more developed root system. As discussed before, isolates of *P. lilacinum* and *P. chlamydosporia* increased the root mass at least in one occasion (Trial 2) (Figure 3). Taken together, we can confirm the increase in common bean root growth due to some fungal isolates. This could play a role in alleviating the damage caused by nematodes, as previously reported to PGPF [50].

Many studies have documented the successful biological control exerted by nematophagous fungi against plant-parasitic nematodes. For instance, Haaraith et al. [77] tested two *Purpureocillium* isolates (namely “E” and “T”) that suppressed the reproduction of *H. glycines* in soybean. In *Abelmoschus esculentus* ‘Sabz Pari’, *P. lilacinum* effectively reduced the number of galls, egg masses, and *M. incognita* reproduction [78]. Tomato plants colonized by *P. chlamydosporia* was found to infect and colonize galls and egg masses of *M. javanica* [79]. Other authors also reported the decrease of nematode per gram of root by biological control agents, which could be attributed to the various mechanisms of these fungi [5,80]. The production and delivery of leukine toxins, chitinases, proteases, and acetic acids are major factors of *P. lilacinum* suppression against phytonematodes [81,82]. In its turn, *P. chlamydosporia* secretes high amounts of chitinases and proteases and a plethora of secondary metabolites, which are important factors in its parasitism of sedentary nematode eggs and females [83]. The two most promising isolates of *P. chlamydosporia* and *P. lilacinum* selected here displayed strong suppression activities toward the population growth of both nematodes and spider mites in planta. Their probable mechanisms may involve the induction of plant defense mechanisms that can assist bean plants in alleviating the damage caused by either *T. urticae* or *H. glycines* attack.

The interactions between endophytic fungi and their hosts could lead to different outcomes, including growth promotion [84,85], increased nitrogen and carbon exchange [86], and even repellence activity against herbivorous insects, phytonematodes, and phytopathogenic fungi [87,88]. However, induced resistance is a prominent mechanism triggered by these fungi, which could negatively affect plant pathogens and pests [89,90,91]. This phenomenon involves the activation of latent defense mechanisms within plants, which are primarily mediated by salicylic acid and jasmonic acid [79]. The role of such mechanisms in the *H. glycines*/*T. urticae* × common bean interactions must be further elucidated.

## 5. Conclusions

Common bean seeds coated with MS of either *P. lilacinum* or *P. chlamydosporia* considerably reduced the population growth of the two-spotted spider mite *T. urticae* and negatively impacted on the population density of *H. glycines*. Notably, the isolates *P. chlamydosporia* ESALQ5406 and *P. lilacinum* ESALQ2593 provided the best results in suppressing spider mite and cyst nematode populations, which make them the strongest candidates for use in the management of these pests. These fungal isolates to some extent improved the growth of bean plants, and also endophytically colonized different parts of bean plants, although only *P. lilacinum* persisted in the soil. Endophytic fungi holding multifaceted roles in plant protection against phytophagous mites and plant-parasitic nematodes represent a valuable asset to be integrated into the management of *T. urtice* and *H. glycines*. In addition, the use of MS as the inoculum source for seed treatment is an innovative strategy that successfully promotes endophytic fungal growth within the host plant, which in turn can minimize their exposure to inconvenient abiotic factors towards a more persistent and cost-effective biocontrol agent. Fungal MS of *P. chlamydosporia* and *P. lilacinum* can be easily mass-produced and scaled-up through submerged liquid fermentation technology, thereby supporting the application of MS in seed treatment with fungal endophytes.

Our findings highlight a new perspective for applying nematophagous fungi to seed treatment by using a robust resting fungal propagule named miscrosclerotia (MS), which can be largely and rapidly produced by submerged liquid fermentation, thus promoting enhanced plant protection and health against below- and above-ground invertebrate pests.

## Figures and Tables

**Figure 1 jof-08-00511-f001:**
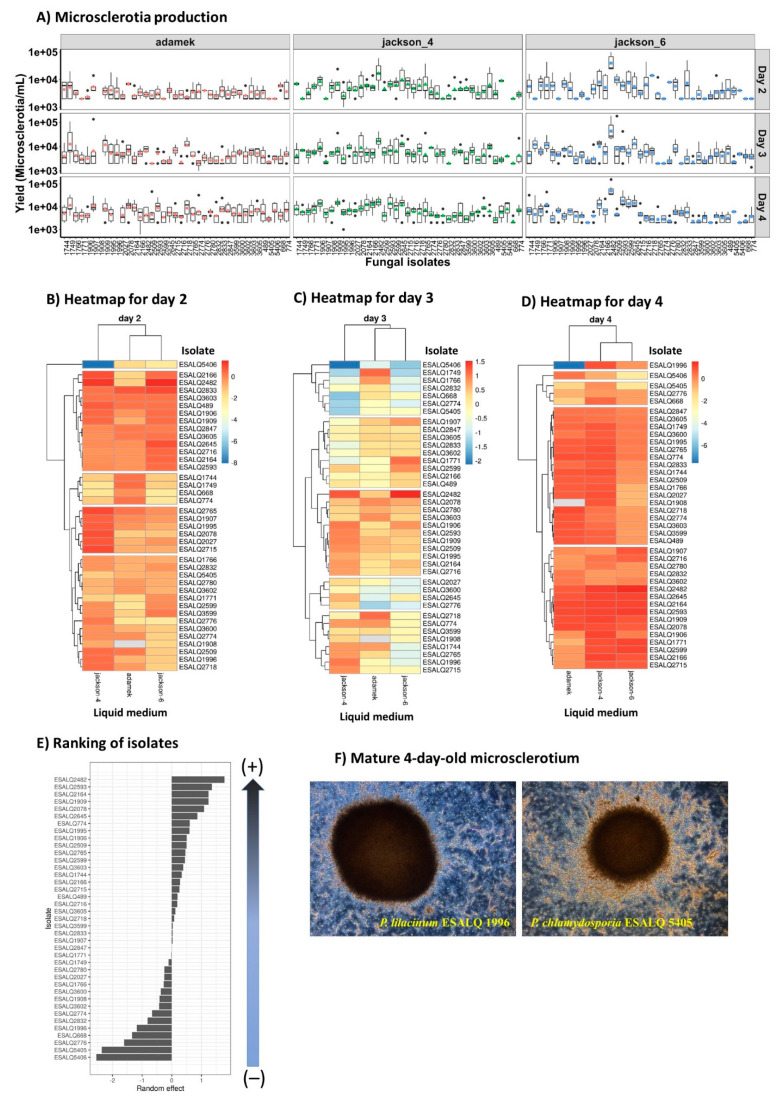
Boxplots (colored symbols are mean values and bold horizontal lines are medians for fungal isolates within each liquid medium) and heatmaps representing the estimated means of microsclerotia (MS) production across the 42 fungal isolates of *Pochonia chlamydosporia* and *Purpureocillium lilacinum* grown in three different liquid media (modified Adamék, Jackson-4, and Jackson-6), from day 2 to 4 (**A**–**D**, respectively). The ranking graph depicting a gradient from strongest to weakest fungal isolate producers based on the estimated general mean of MS concentration (**E**). The scale from blue to red means that high MS are highlighted in red, whereas low MS yields are assigned with blue. Typical 4-day-old liquid-grown MS of *P. lilacinum* ESALQ1996 and *P. chlamydosporia* ESALQ5405 (**F**).

**Figure 2 jof-08-00511-f002:**
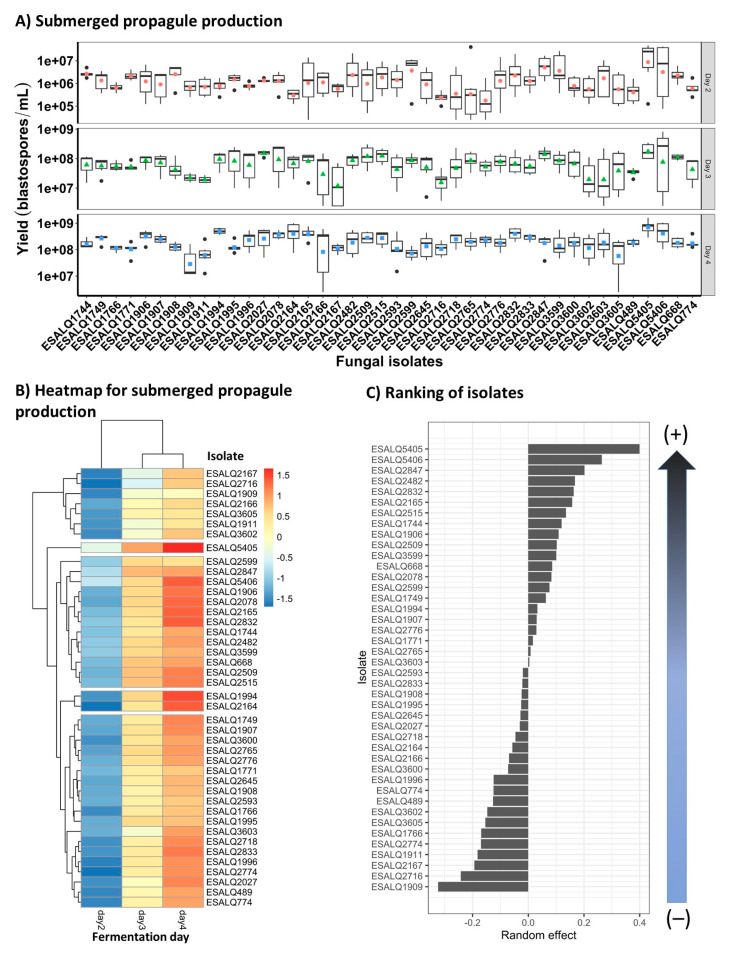
Boxplots (colored symbols are mean values and bold horizontal lines are medians for fungal isolates within each fermentation day) and heatmap and a graph rank depicting a gradient between strongest and weakest fungal isolate producers for predicted mean concentration of submerged propagules (blastospores) of *Purpureocillium lilacinum* and *Pochonia chlamydosporia* on the fourth day of cultivation in the modified Adamék medium (**A**,**B**, respectively). Scale from blue to red means that high yields of submerged propagules are highlighted in red, whereas low yields are assigned with blue (**C**).

**Figure 3 jof-08-00511-f003:**
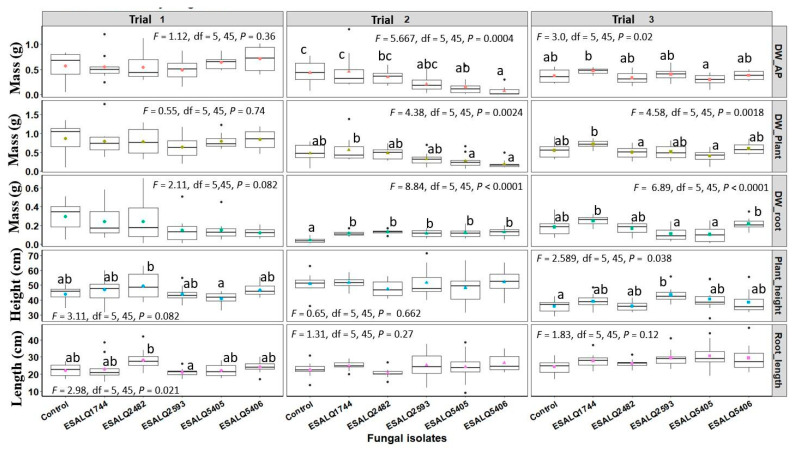
Common bean (*Phaseolus vulgaris*) plant traits (dry weight of the aerial part, total plant dry weight, root dry weight, plant height, root length) after seed treatment with different isolates of *Purpureocillium lilacinum* (ESALQ1744, ESALQ2482, and ESALQ2593) and *Pochonia chlamydosporia* (ESALQ5405 and ESALQ5406) at 45 days after fungal inoculation and seeding, under greenhouse conditions. Colored symbols inside boxplots represent means (±S.E., *n* = 10 plants per experiment and treatment), and treatments with no letters in common are statistically different at *p* < 0.05 based on Tukey HSD. Plant traits evaluated: dry weight of aerial part (g, DW_AP), dry weight of whole plant (g, DW_Plant), dry weight of roots (g, DW_root), plant height (cm), and root length (cm).

**Figure 4 jof-08-00511-f004:**
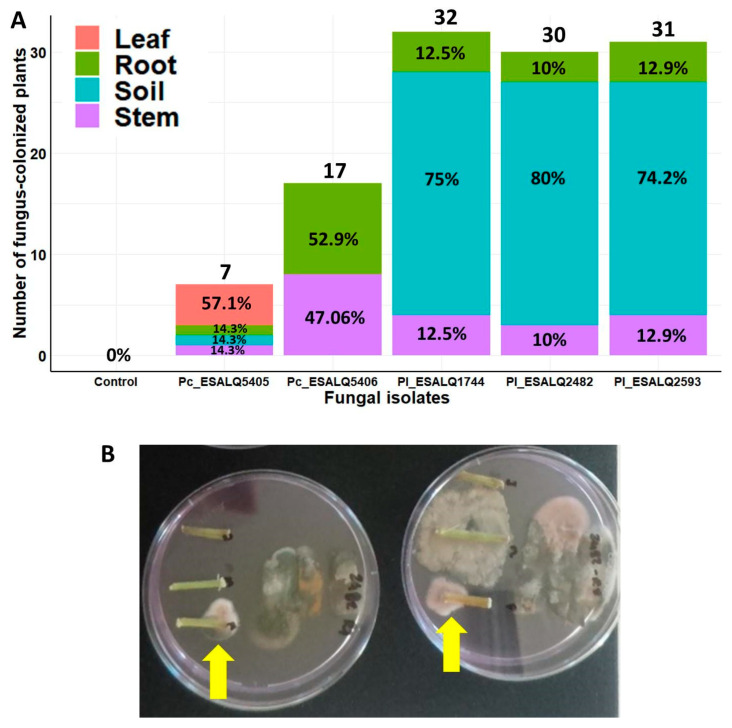
Spatial distribution and frequency of endophytic colonization after seed treatment with *Purpureocillium lilacinum* (Pl) and *Pochonia chlamydosporia* (Pc) within different plant tissues from potted bean plants and from the potting soil after 45 days of fungal inoculation and seeding under greenhouse conditions (**A**). Selective medium showing bean stem fragments endophytically colonized by *P. lilacinum* (indicated by yellow arrows) in (**B**). The total number of plants evaluated per treatment: *n* = 3 experiments × 4 structures × 10 repetitions = 120 plants.

**Figure 5 jof-08-00511-f005:**
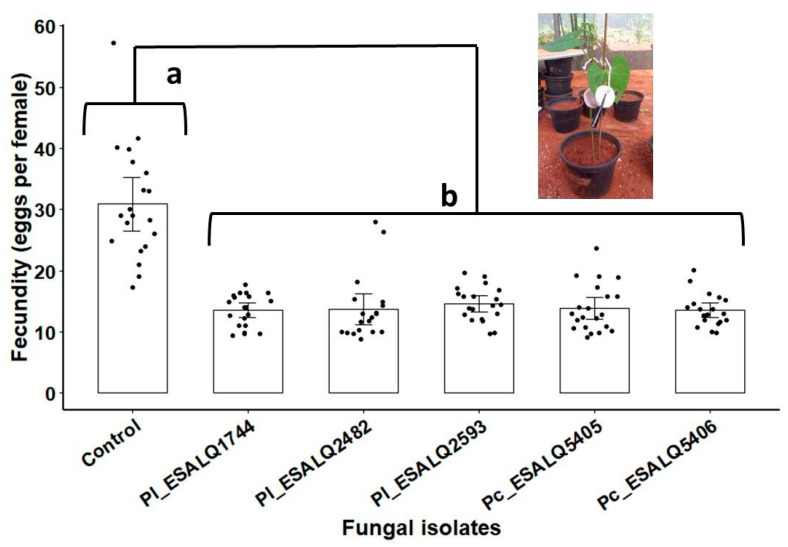
The number of eggs laid by *Tetranychus urticae* after seven days of infestation in bean plants derived from seeds inoculated with six different treatments: fungus-free control (0.5% *w*/*v* of Arabic gum) and fungal treatments based on seed-coating with MS (dose equivalent 10^12^ conidia ha^−1^) of the isolates ESALQ1744, ESALQ2482 and ESALQ2593 of *Purpureocillium lilacinum* (Pl), and isolates ESALQ5405 and ESALQ5406 of *Pochonia chlamydosporia* (Pc). Means (± 95% confidence interval, *n* = 20 biological replicates from two independent experiments) of spider mite counts (eggs per leaf arena). Treatments (vertical bars) followed by distinct letters differ significantly at *p* < 0.05 based on Tukey HSD.

**Figure 6 jof-08-00511-f006:**
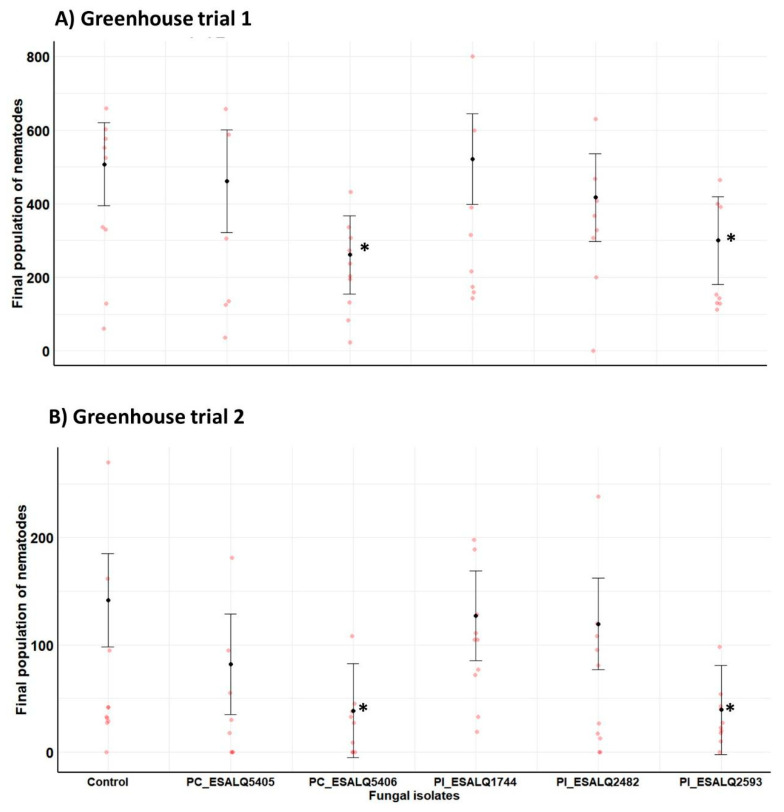
Impact of *Purpureocillium lilacinum* (Pl) and *Pochonia chlamydosporia* (PC) isolates on the final population of the soybean cyst nematode *Heterodera glycines* in total fresh root mass from potted bean plants at 45 days after fungal inoculation and seeding under greenhouse conditions. Means with 95% confidence intervals (*n* = 10 plants per treatment in red symbols) followed by asterisks (*) are significantly different from untreated control plants infested only with the cyst nematode (Likelihood ratio test at *p* < 0.05).

**Figure 7 jof-08-00511-f007:**
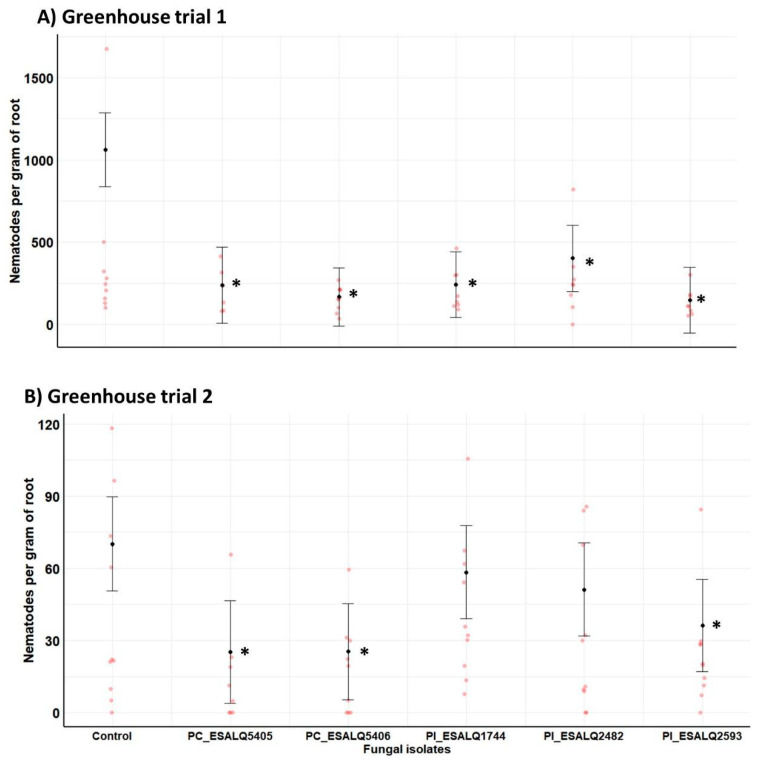
Impact of *Purpureocillium lilacinum* (Pl) and *Pochonia chlamydosporia* (PC) isolates on the population density (nematodes per gram of fresh root mass) of the soybean cyst nematode *Heterodera glycines* from potted bean plants at 45 days after fungal inoculation and seeding, under greenhouse conditions. Means with 95% confidence intervals (*n* = 10 plants per treatment in red symbol) followed by asterisks (*) are significantly different from untreated control plants infested only with the nematode (Likelihood ratio test at *p* < 0.05).

**Table 1 jof-08-00511-t001:** Description of the origin and collection date of *Purpureocillium lilacinum* and *Pochonia chlamydosporia* isolates used in the experiments.

Isolate Code	Collection Date	Species	Origin	
			Host/Substrate	Location/Biome
ESALQ489	12/05/1986	*Purpureocillium lilacinum*	Lepidptera: Hemileucidae	Teixeira de Freitas—BA
ESALQ668	10/21/1987	*Purpureocillium lilacinum*	*Solenopsis* sp.	Porto Feliz—SP
ESALQ774	03/01/1988	*Purpureocillium lilacinum*	*Solenopsis* sp. nest soil	Rio Grande do Sul
ESALQ1744	08/13/2012	*Purpureocillium lilacinum*	Soil: Selective medium	Caatinga—Palma
ESALQ1749	08/13/2012	*Purpureocillium lilacinum*	Soil: Selective medium	Caatinga
ESALQ1766	03/27/2012	*Purpureocillium lilacinum*	Soil: Selective medium	Savanna
ESALQ1771	03/27/2012	*Purpureocillium lilacinum*	Soil: Selective medium	Savanna
ESALQ1906	03/06/2012	*Purpureocillium lilacinum*	Soil: Insect bait	Sinop—MT
ESALQ1907	03/06/2012	*Purpureocillium lilacinum*	Soil: Insect bait	Sinop—MT
ESALQ1908	03/06/2012	*Purpureocillium lilacinum*	Soil: Insect bait	Sinop—MT
ESALQ1909	03/06/2012	*Purpureocillium lilacinum*	Soil: Insect bait	Sinop—MT
ESALQ1994	03/14/2012	*Purpureocillium lilacinum*	Soil: Insect bait	Teotônio Vilela—AL
ESALQ1995	03/21/2012	*Purpureocillium lilacinum*	Soil: Selective medium	Delmiro Gouveia—AL
ESALQ1996	03/14/2012	*Purpureocillium lilacinum*	Soil: Insect bait	Teotônio Vilela—AL
ESALQ2077	03/27/2012	*Purpureocillium lilacinum*	Soil: Insect bait	Rio Verde—GO
ESALQ2078	03/27/2012	*Purpureocillium lilacinum*	Soil: Insect bait	Rio Verde—GO
ESALQ2164	08/21/2012	*Purpureocillium lilacinum*	Soil: Selective medium	Delmiro Gouveia—AL
ESALQ2165	08/21/2012	*Purpureocillium lilacinum*	Soil: Selective medium	Delmiro Gouveia—AL
ESALQ2166	08/21/2012	*Purpureocillium lilacinum*	Soil: Selective medium	Delmiro Gouveia—AL
ESALQ2167	08/21/2012	*Purpureocillium lilacinum*	Soil: Selective medium	Delmiro Gouveia—AL
ESALQ2482	09/05/2012	*Purpureocillium lilacinum*	Soil: Insect bait	Rio Verde—GO
ESALQ2509	06/26/2012	*Purpureocillium lilacinum*	Soil: Insect bait	Aceguá-RS
ESALQ2593	07/31/2012	*Purpureocillium lilacinum*	Soil: Selective medium	Amazon
ESALQ2599	09/04/2012	*Purpureocillium lilacinum*	Soil: Selective medium	Savanna
ESALQ2645	07/31/2012	*Purpureocillium lilacinum*	Soil: Selective medium	Amazon
ESALQ2715	09/04/2012	*Purpureocillium lilacinum*	Soil: Selective medium	Savanna
ESALQ2716	03/21/2012	*Purpureocillium lilacinum*	Soil: Selective medium	Caatinga
ESALQ2718	10/03/2012	*Purpureocillium lilacinum*	Soil: Selective medium	Savanna
ESALQ2765	06/26/2012	*Purpureocillium lilacinum*	Soil: Selective medium	Pampa
ESALQ2774	08/14/2012	*Purpureocillium lilacinum*	Soil: Selective medium	Sinop—MT
ESALQ2776	08/14/2012	*Purpureocillium lilacinum*	Soil: Selective medium	Sinop—MT
ESALQ2780	08/14/2012	*Purpureocillium lilacinum*	Soil: Selective medium	Teotônio Vilela—AL
ESALQ2832	10/03/2012	*Purpureocillium lilacinum*	Soil: Selective medium	Rio Verde—GO
ESALQ2833	10/03/2012	*Purpureocillium lilacinum*	Soil: Selective medium	Rio Verde—GO
ESALQ2847	03/14/2012	*Purpureocillium lilacinum*	Soil: Selective medium	Atlantic florest
ESALQ3599	02/19/2013	*Purpureocillium lilacinum*	*Toxoptera citricida*	Itirapina—SP
ESALQ3600	02/19/2013	*Purpureocillium lilacinum*	Soil	Conchal—SP
ESALQ3602	02/19/2013	*Purpureocillium lilacinum*	Soil	Conchal—SP
ESALQ3603	02/19/2013	*Purpureocillium lilacinum*	Soil	Conchal—SP
ESALQ3605	04/03/2013	*Purpureocillium lilacinum*	Soil	Nova Europa—SP
ESALQ5405	03/27/2012	*Pochonia chlamydosporia*	Soil: Selective medium	Rio Verde—GO
ESALQ5406	03/27/2012	*Pochonia chlamydosporia*	Soil: Selective medium	Rio Verde—GO

## Data Availability

Not applicable.

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
