# Peer review of "Production of Purpureocillium lilacinum and Pochonia chlamydosporia by Submerged Liquid Fermentation and Bioactivity against Tetranychus urticae and Heterodera glycines through Seed Inoculation"

_jof, 2022, doi:10.3390/jof8050511_

Round 1

Reviewer 1 Report

This submitted manuscript illustrates about the production of Purpureocillium lilacinum and Pochonia chlamydosporia by submerged liquid fermentation and their bioactivity against Tetranychus urticae and Heterodera glycines through seed inoculation.

In brief, this paper provides a comprehensive study of the effect of different liquid media on the growth of 42 isolates of P. lilacinum and two of P. chlamydosporia and the effect of two of the most promising isolates on growth promotion and the control of the two-spotted spider mite (Tetranychus urticae) and the soybean cyst nematode (Heterodera glycines).

Overall, this paper is innovative, well-written and has merits to be published in the Journal of Fungi as it provides new insights for the applicability of fungal microsclerotia in seed treatment to control Phaseolus vulgaris pests

However, there are some observations before this manuscript could be considered to be published in its current form and my specific comments are appended below:

There was no line number which usually makes easier for the reviewer to make specific comments

Introduction: The introduction is somewhat bigger; the author could merge some of the paragraphs to make finally the introduction into 4-5 paragraphs.

Methods: in the section 2.4, it is not mentioned whether the author sterilized the substrate or potting mixture (soil and sand) before the treated seeds were sown which could affect the colonization process.

It appears that the author tested the colonization after the end of the trial (45 days after sowing), whereas, in the section 3.5 (Effects of nematophagous fungi on population growth of T. urticae) they did the study at 21 days after inoculation or seeding. Now after 21 days, there is no information or data whether the author did any test about the colonization of the plants used in the insect study.

3.5. Effects of nematophagous fungi on population growth of T. urticae: the scientific name should be in the italic form

3.6. Effects of nematophagous fungi on population growth of H. glycines: the scientific name should be in the italic form.

Discussion: in the discussion section the authors should improve the insect related part with more ref (now there are papers with fungal endophyte, T. urticae interactions)

Author Response

Response to Reviewer 1

Comments and Suggestions for Authors

This submitted manuscript illustrates about the production of Purpureocillium lilacinum and Pochonia chlamydosporia by submerged liquid fermentation and their bioactivity against Tetranychus urticae and Heterodera glycines through seed inoculation.

In brief, this paper provides a comprehensive study of the effect of different liquid media on the growth of 42 isolates of P. lilacinum and two of P. chlamydosporia and the effect of two of the most promising isolates on growth promotion and the control of the two-spotted spider mite (Tetranychus urticae) and the soybean cyst nematode (Heterodera glycines).

Overall, this paper is innovative, well-written and has merits to be published in the Journal of Fungi as it provides new insights for the applicability of fungal microsclerotia in seed treatment to control Phaseolus vulgaris pests

However, there are some observations before this manuscript could be considered to be published in its current form and my specific comments are appended below:

There was no line number which usually makes easier for the reviewer to make specific comments

Response: Sorry, we thought the journal system would automatically add line numbering once the PDF was generated. We now have added line numbers to this revised version of our manuscript.

Introduction: The introduction is somewhat bigger; the author could merge some of the paragraphs to make finally the introduction into 4-5 paragraphs.

Response: Thanks for the hint. We reduced the introduction to 8 paragraphs without omitting important information in the context of our study.

Methods: in the section 2.4, it is not mentioned whether the author sterilized the substrate or potting mixture (soil and sand) before the treated seeds were sown which could affect the colonization process.

Response: The potting mixture (soil+sand) was not autoclaved (sterilized), so it was natural in order to simulate a real soil environment. This information was added in the Materials and Methods. Please, see line 135.

It appears that the author tested the colonization after the end of the trial (45 days after sowing), whereas, in the section 3.5 (Effects of nematophagous fungi on population growth of T. urticae) they did the study at 21 days after inoculation or seeding. Now after 21 days, there is no information or data whether the author did any test about the colonization of the plants used in the insect study.

Response: Correct. We did not evaluate the fungal colonization at 21 days after inoculation and seeding. The reason for this experiment was to investigate the impact of fungal endophytes in suppressing the spider mite female's fecundity, according to the protocol of Canassa et al. (2019 [reference 44]). The other experiment was conducted for 45 days post-seeding to measure the endophytism successful rate of nematophagous fungi. Thus, we set up two independent experiments, one for spider mite and another to evaluate fungal endophytism on plant growth and development.

3.5. Effects of nematophagous fungi on population growth of T. urticae: the scientific name should be in the italic form.

Response: Thanks, done.

3.6. Effects of nematophagous fungi on population growth of H. glycines: the scientific name should be in the italic form.

Response: Thanks, done.

Discussion: in the discussion section the authors should improve the insect related part with more ref (now there are papers with fungal endophyte, T. urticae interactions)

Response: We have improved the T. urticae part in our discussion using the following references:

Al Khoury C. Molecular insight into the endophytic growth of Beauveria bassiana within Phaseolus vulgaris in the presence or absence of Tetranychus urticae. Mol Biol Rep. 2021 Mar;48(3):2485-2496. doi: 10.1007/s11033-021-06283-3

L.R. Jaber, B.H. Ownley. Can we use entomopathogenic fungi as endophytes for dual biological control of insect pests and plant pathogens? Biol. Control, 116 (2018), pp. 36-45

Dash CK, Bamisile BS, Keppanan R, Qasim M, Lin YW, Ul Islam S, Hussain M, Wang LD. 2018. Endophytic entomopathogenic fungi enhance the growth of phaseolus vulgaris L. (fabaceae) and negatively affect the development and reproduction of tetranychus urticae koch (acari: tetranychidae). Microb Pathog. Dec; 125:385–392. doi:10.1016/j.micpath.2018.09.044.

Canassa, F.; Tall, S.; Moral, R. A.; Lara, I. A. R.; Delalibera, I.; Meyling, N. V. Effects of bean seed treatment by the entomopathogenic fungi Metarhizium robertsii and Beauveria bassiana on plant growth, spider mite population and behavior of predatory mites. Bio Control, 2019, volume 132, pp. 199-208. https://doi.org/10.1016/j.biocontrol.2019.02.003

Changes were made in lines 811 to 827 in the revised version of this manuscript.

Reviewer 2 Report

General Comments

Without line numbers, it is rather impossible for me to point out line by line the changes that are required to be made in the manuscript. For ease of read and review, I assume it is imperative to include line numbers when submitting a manuscript even if this was not made mandatory by the journal. I have highlighted the errors to be corrected in the attached manuscript .pdf file, and I hope the authors would be able to find, and make amends as appropriate in order to improve the manuscript.

Specific Comments

Abstract

Here, so many short statements that could have been linked with conjunctions, such as ‘whereas’, ‘while’, ‘on the other hand’ etc. in order to enhance readability. In addition, after a ‘.’, it’s ideal to write ‘P’ in full, as in ‘Purpureocillium

The entire statement should be reworded, viz.  – P. chlamydosporia ESALQ5406 and P. lilacinum ESALQ2593 decreased cyst nematode population. P. lilacinum was more detected in soil, whereas P. chlamydosporia colonized all plant parts. P. chlamydosporia ESALQ5406 improved root development.

‘Suitabily?’ perhaps the authors mean to write ‘suitability’. Please, replace ‘Suitabily’ with ‘possibility’ and revise the statement as ‘These findings demonstrate the possibility of producing submerged propagules of P. chlamydosporia and P. lilacinum by liquid culture…

1. Introduction

…are currently used as microbial pesticides against plant-parasitic nematodes in Brazil [5, 6].

P…G…P should be lowercase letters in …plants playing an essential role in plant growth promoting.

Replace 'scarcity' with a more appropriate synonym in… “In that sense, the scarcity of studies…”

2. Materials and Methods

Delete the year; (2018) (2015a) (1965) from - developed by Iwanicki et al. (2018) [40] by Mascarin et al. (2015a) [21] and by Adamék (1965) [41].

Also here, remove (2019) - proposed by Machado and Silva (2019) [46].

Revise as – ‘Prior to the inoculation, the nematode suspension…’

Here again, no need to add the year - Coolen and D’Herde (1972) [48].

Please, correct this error in the entire manuscript. It is important to stick with the journal referencing style (MDPI ref-style) all through the manuscript.

Amend the referencing error here also - Goodness-of-fit for model selection was assessed using half-normal plots with a simulated envelope ([52] Moral et al., 2017) and wormplots [51] (Stasinopoulos & Rigby, 2007). All analyses were carried out using R [53].

3. Results

…Increase in root mass was obtained in trial 2

… The endophytic ability of these nematophagous fungal isolates in potted bean plants.

The following statement is not grammatically correct, please, amend. Viz. …were able to different extents endophytically colonize different parts of bean plants…

…but only isolate P. chlamydosporia ESALQ5405 was

The following statement is not entirely correct - Actually, untreated plants and P. lilacinum ESALQ1744 treated plants were conducive to increasing the nematode population in both trials.

Isolate ESALQ1744 might have been unsuccessful in repelling nematode reproduction, however, I assume it is misleading to claim the isolate increased the nematode population.

4. Discussion

Song et al. [61] were the first to describe…

…most of the microbial biopesticides registered in Brazil in 2021 have been targeted at controlling plant-parasitic nematodes [7]

…nutriotional requirements should be nutritional requirements

Song et al. [64] observed that when culturing Metarhizium rileyi in nitrogen and carbon-deficient

Romero-Rangel et al. [73] obtained 3.8 × 107 blastospores ml-1

Jackson et al. [74] reported the production of…

Mascarin et al. [21, 71].

Add a ‘,’ after ESALQ2482 viz. – ‘P. lilacinum ESALQ1744, ESALQ2482, and ESALQ2593,’

Corroborating this result, Canassa et al. [45] reported suppression in the…

For instance, Haaraith et al. [85] tested two Purpureocillium isolates…

Author Response

Response to Reviewer 2

General Comments

Without line numbers, it is rather impossible for me to point out line by line the changes that are required to be made in the manuscript. For ease of read and review, I assume it is imperative to include line numbers when submitting a manuscript even if this was not made mandatory by the journal. I have highlighted the errors to be corrected in the attached manuscript .pdf file, and I hope the authors would be able to find, and make amends as appropriate in order to improve the manuscript.

Response: Thank you. All comments were valuable to improving our manuscript, and they were considered in our revised version of this manuscript. All changes made throughout the manuscript text are marked in red.

Specific Comments

Abstract

Here, so many short statements that could have been linked with conjunctions, such as ‘whereas’, ‘while’, ‘on the other hand’ etc. in order to enhance readability. In addition, after a ‘.’, it’s ideal to write ‘P’ in full, as in ‘Purpureocillium

Response: Ok, thank you for these hints. We followed your suggestions and improved readability by using conjunctions between sentences.

The entire statement should be reworded, viz.  – P. chlamydosporia ESALQ5406 and P. lilacinum ESALQ2593 decreased cyst nematode population. P. lilacinum was more detected in soil, whereas P. chlamydosporia colonized all plant parts. P. chlamydosporia ESALQ5406 improved root development.

Response: OK, these sentences were reworded accordingly, and now it reads like:

“All fungal isolates reduced the T. urticae fecundity in inoculated plants through seed treatment, while P. chlamydosporia ESALQ5406 and P. lilacinum ESALQ2593 decreased cyst nematode population. Purpureocillium lilacinum was more frequently detected in soil, whereas P. chlamydosporia colonized all plant parts. Pochonia chlamydosporia ESALQ5406 improved root development of bean plants.”

‘Suitabily?’ perhaps the authors mean to write ‘suitability’. Please, replace ‘Suitabily’ with ‘possibility’ and revise the statement as ‘These findings demonstrate the possibility of producing submerged propagules of P. chlamydosporia and P. lilacinum by liquid culture…

Response: Thanks, the sentence was revised as suggested. It now reads, “These findings demonstrate the possibility of producing submerged propagules of P. ch….” (lines 34-37)

  1. Introduction

…are currently used as microbial pesticides against plant-parasitic nematodes in Brazil [5, 6].

Response: Changed as suggested. Thanks. (line 59)

P…G…P should be lowercase letters in …plants playing an essential role in plant growth promoting.

Response: Ok, changed as suggested. (line 68)

Replace 'scarcity' with a more appropriate synonym in… “In that sense, the scarcity of studies…”

 Response: Ok, ‘scarcity’ was replaced by ‘limited’, and now it reads like “there are limited studies..”. (line 168)

  1. Materials and Methods

Delete the year; (2018) (2015a) (1965) from - developed by Iwanicki et al. (2018) [40] by Mascarin et al. (2015a) [21] and by Adamék (1965) [41].

Response: Done. Thanks.

Also here, remove (2019) - proposed by Machado and Silva (2019) [46].

Response: Done. Thanks.

Revise as – ‘Prior to the inoculation, the nematode suspension…’

Response: Done. Thanks.

Here again, no need to add the year - Coolen and D’Herde (1972) [48].

Response: Done. Thanks.

Please, correct this error in the entire manuscript. It is important to stick with the journal referencing style (MDPI ref-style) all through the manuscript.

Response: Done. Thanks.

Amend the referencing error here also - Goodness-of-fit for model selection was assessed using half-normal plots with a simulated envelope ([52] Moral et al., 2017) and wormplots [51] (Stasinopoulos & Rigby, 2007). All analyses were carried out using R [53].

Response: Done. Thanks.

  1. Results

…Increase in root mass was obtained in trial 2

Response: Done. Thanks. (line 624)

… The endophytic ability of these nematophagous fungal isolates in potted bean plants.

Response: Done. Thanks. (line 652)

The following statement is not grammatically correct, please, amend. Viz. …were able to different extents endophytically colonize different parts of bean plants…

Response: Sentence grammatically corrected to “…were able at different levels to endophytically...” (line 660)

…but only isolate P. chlamydosporia ESALQ5405 was

Response: Done. Thanks. (line 662)

The following statement is not entirely correct - Actually, untreated plants and P. lilacinum ESALQ1744 treated plants were conducive to increasing the nematode population in both trials.

Isolate ESALQ1744 might have been unsuccessful in repelling nematode reproduction, however, I assume it is misleading to claim the isolate increased the nematode population.

Response: This sentence was amended to “Actually, untreated plants had an increase in population growth of H. glycines, while P. lilacinum ESALQ1744 were unsuccessful in reducing nematode reproduction in both trials”. (lines 729-731)

  1. Discussion

Song et al. [61] were the first to describe…

Response: Done, thanks.

…most of the microbial biopesticides registered in Brazil in 2021 have been targeted at controlling plant-parasitic nematodes [7]

Response:  Sentence amended to “…have been targeted at controlling plant…” (line 813)

…nutriotional requirements should be nutritional requirements

Response: Corrected, as suggested. Thanks.

Song et al. [64] observed that when culturing Metarhizium rileyi in nitrogen and carbon-deficient

Response: Done, thanks.

Romero-Rangel et al. [73] obtained 3.8 × 107 blastospores ml-1

Response: Done, thanks.

Jackson et al. [74] reported the production of…

Response: Done, thanks.

Mascarin et al. [21, 71].

Response: Done, thanks.

Add a ‘,’ after ESALQ2482 viz. – ‘P. lilacinum ESALQ1744, ESALQ2482, and ESALQ2593,’

Response: Done, thanks.

Corroborating this result, Canassa et al. [45] reported suppression in the…

Response: Done, thanks.

For instance, Haaraith et al. [85] tested two Purpureocillium isolates…

Response: Done, thanks.

All comments made by reviewer #2 in the PDF file of our manuscript were considered in the new revised file of our manuscript, where all changes were marked in red color. Thank you very much for these valuable inputs.
